# GPR88 localization to primary cilia in neurons is cell-type specific

Yenni H Li Guan[1], Brigitte L Kieffer[2], Mark von Zastrow[1], Aliza T Ehrlich[1]

**GPR88 is an orphan G protein-coupled receptor that regulates dopamine neurotransmission and is a target for neuropsychiatric disorders. In addition to the somatic membrane, GPR88 can localize to the primary cilium, a membrane microdomain known for dynamically enriching receptors and signaling molecules. However, the distribution of GPR88 in neuronal primary cilia remains uncharacterized. Here, we characterize GPR88 distribution at primary cilia in two brain areas. We show that in the striatum, GPR88 localizes both to somatodendritic and primary cilia compartments on inhibitory GABAergic medium spiny neurons. In contrast, in the somatosensory cortex, GPR88 localizes to somatodendritic and nuclear compartments and not primary cilia of excitatory spiny stellate neurons. In addition, we found that cilia density and length were similar between _Gpr88_ knockout and wild-type animals. Together, we provide key evidence for neuronal cell-type specific regulation of GPR88 localization to primary cilia, suggesting neuron subtype specific regulatory mechanisms govern receptor ciliary targeting in the brain.**

## Introduction

G protein-coupled receptors (GPCRs) are the largest group of membrane proteins and are involved in critical physiological functions including neuronal function (1, 2). A subset of GPCRs involved in neuromodulation has been found to be targeted to primary cilia (3, 4). Primary cilia are non-motile microtubule-based organelles that protrude off of the cell surface of most cell types in the human body, including neurons throughout the brain (5, 6, 7). In the developing brain, primary cilia are critical locations for Sonic hedgehog (Shh) and Wnt signaling pathways and defects in cilia formation or function can lead to severe disorders known as "ciliopathies" (8, 9). In the mature brain, the function of primary cilia and the GPCRs that enrich it is an emerging area.

Recent work suggests that GPCRs localized to mature primary cilia could be critical for neuromodulation processes including excitatory synaptic regulation (10) axo-ciliary synaptic regulation of nuclear signaling (11, 12), and energy homeostasis (13, 14). Whereas prototypical ciliary neuromodulatory GPCRs (5HTR6, SSTR3) have been reported to be highly enriched at neuronal primary cilia (15, 16), other ciliary GPCRs lack efficient immuno-detection methods to assess ciliary localization in physiological systems. Furthermore, GPCR localization to neuronal primary cilia may be specifically regulated by other proteins or dynamically occur in response to signaling molecules. For example, the dopamine receptor DRD1 is only visible at the primary cilium in brain sections of mice lacking ciliary exit machinery, Bardet-Biedl syndrome proteins (17), despite being concentrated there in heterologous systems (18). In addition to, the melanocortin 4 receptor requires an accessory protein to localize to primary cilia that may not be present in heterologous cell systems (19). Meanwhile the mu opioid receptor is visible at primary cilia in the habenula of WT mice but requires an extension of its carboxyl terminus to be transported to primary cilia in heterologous cell culture systems (20). Thus, to what extent neuromodulatory receptors localize to primary cilia in physiological systems and how GPCR localization to primary cilia is regulated is not clear.

We previously showed that GPR88, an orphan receptor with high interest as a neurotherapeutic target for addiction (21, 22), localizes to primary cilia when overexpressed in ciliated mouse kidney cells and rat striatal neurons (23) but in physiological systems GPR88 displays time-dependent ciliary enrichment in the nascent period and regional selectivity in the mature brain (24). The time-dependent nature of GPR88 localization to primary cilia combined with loss of function effects on motor and intellectual ability (25, 26), suggests a role for the receptor in cilia morphology. In contrast the regional selectivity of GPR88 localization to primary cilia suggests that cilia localization of GPR88 may be neuronal context specific.

Here, we test the above hypotheses by first systematically identifying neuronal subtypes in the striatum and the somatosensory cortex, two areas with the highest GPR88 expression (24, 27), using GPR88-Venus knock-in mice (24) and second through

---

[1]Department of Psychiatry and Behavioral Sciences, University of California, San Francisco, San Francisco, CA, USA   [2]INSERM UMR-S 1329, Strasbourg Translational Neuroscience and Psychiatry, Centre de Recherche en Biomédecine de Strasbourg, Université de Strasbourg (UNISTRA), Strasbourg, France

Correspondence: Aliza.ehrlich@ucsf.edu

detailed quantification of neuronal primary cilia density and length in *Gpr88*-knockout (KO) mice (25). We find that GPR88 is restricted to somatodendritic and nuclear but not primary cilia compartments on SATB2+ excitatory neurons in the cortex and to somatodendritic and primary cilia compartments on DARPP-32+ medium spiny neurons (MSNs) in the striatum suggesting that GPR88 localization to primary cilia is dependent on the cell subtype. In addition, we find that GPR88 loss of function does not perturb cilia density or length. These findings suggest that GPR88 loss of function behavioral effects may arise from perturbed GPR88 signaling rather than primary cilia morphology defects. Our results support the concept that ciliary GPCRs require a specific cellular context to be localized to primary cilia.

# Results

### Expression and quantification of cortical and striatal neuronal subtypes

To investigate the overarching hypothesis that GPR88 localization to primary cilia is regulated by neuronal subtypes, we first identified and quantified the expression patterns of reported neuronal subtype markers in both somatosensory cortex layer 4 (SSCtx L4) and dorsal striatum (dStr) (28, 29, 30, 31) in GPR88 WT (*GPR88*^WT/WT) and GPR88-Venus (*GPR88*^Venus/Venus) animals. We focused on these two brain regions because the striatum and cortex show the highest levels of GPR88 expression (24, 27). Functional MRI studies further implicate these regions in the primary functions of GPR88, as deletion of the receptor significantly impacts their activity (32). We identified three neuronal subtypes in the cortex and two neuronal subtypes in the striatum that could reliably be immunostained with commercial antibodies and were representative of the major neuronal subtypes found in these brain regions (29, 30, 33, 34, 35, 36, 37, 38). Specifically, we assessed parvalbumin (PV), somatostatin (SST), and special AT-rich sequence-binding protein 2 (SATB2) in the cortex. In the striatum, we assessed two neuronal subtype populations choline acetyltransferase (CHAT) and dopamine and cAMP regulated phosphoprotein (DARPP-32).

We first focused on identifying the neuronal subtypes in the SSCtx L4 (Fig 1A). Previous reports have shown that PV and SST are the most abundant inhibitory neuronal markers in SSCtx (29, 34). To label excitatory neurons in SSCtx, SATB2 was selected due to its exclusive expression in this subtype (39). With confocal imaging, we observed that PV (Fig 1B) and SST (Fig 1C) expressing neurons were sparsely distributed across the cortical region while SATB2 (Fig 1D) expression was denser and more widely distributed. Neuronal quantification was carried out by counting the total number of each neuronal subtype and obtaining the percentage of neuronal subtype to the total counted neurons in the SSCtx L4. Inhibitory PV subtype neurons made up 3.22% ± 0.5% *GPR88*^WT/WT and 5.07% ± 0.4% *GPR88*^Venus/Venus of total neurons in the SSCtx L4 with the latter having significantly higher PV neurons (Fig 1B). The SST inhibitory subtype neurons made up a minor population of 5.08% ± 1.1% and 2.5% ± 0.3% of total neurons for *GPR88*^WT/WT and *GPR88*^Venus/Venus mice, respectively, with no statistical significance

(Fig 1C). The largest population of cortical neurons was the SATB2 excitatory neurons which were not significantly different between genotypes with 91.7% ± 1.5% and 92.41% ± 0.7% of SATB2+ neurons for *GPR88*^WT/WT and *GPR88*^Venus/Venus mice, respectively (Fig 1D). Thus, the distributions of cortical neuronal populations across the two genotypes were similar with the majority being excitatory SATB2+ and the minority being inhibitory PV+ and SST+ neurons.

Next, we evaluated neuronal subtypes in the dorsal striatum (Fig 1E). The striatum is predominantly composed of GABAergic MSNs (31), and contains sparse distribution of cholinergic interneurons (28). To distinguish MSNs, we used anti-DARPP-32 as a specific marker, whereas anti-CHAT was used to label cholinergic interneurons. We observed sparse labeling of CHAT+ cholinergic interneurons and dense widespread labeling of DARPP-32+ neurons in the striatum (Fig 1F and G). Neuronal quantification shows CHAT+ neurons were sparsely distributed in both genotypes accounting for only 3.91% ± 0.8% and 1.4% ± 0.1% of total neurons for *GPR88*^WT/WT and *GPR88*^Venus/Venus mice, respectively, with *GPR88*^WT/WT having significantly higher percentage of cholinergic interneurons (Fig 1F). The DARPP-32+ MSNs were the most abundant with 96.09% ± 0.8% for *GPR88*^WT/WT, and 98.59% ± 0.1% for *GPR88*^Venus/Venus (Fig 1G). A significantly higher percentage of DARPP-32 MSNs were detected in *GPR88*^Venus/Venus as compared with *GPR88*^WT/WT. Collectively the distribution of striatal neuronal subtypes in both genotypes was within reported ranges with greater than 95% of neurons being MSNs and the minority population being cholinergic interneurons.

### Primary cilia are found across neuronal subtypes in the somatosensory cortex and striatum

Neuronal primary cilia are microtubule-based organelles that play a key role in a variety of signaling pathways impacting neuronal excitability and behavior (10, 40). We previously reported the observation that GPR88 localizes to ARL13B labeled primary cilia in the striatum, whereas cortical neurons do not localize GPR88 to primary cilia (24). We reasoned that this could be due to incomplete labeling of neuronal primary cilia with ARL13B which predominantly labels glial cells or the lack of primary cilia on cortical GPR88+ neurons. To test this, we compared different commercially available antibodies for a known neuronal cilia marker, Adenylyl cyclase 3 (AC3) to ARL13B (Fig S1). AC3 broadly labeled neuronal primary cilia in the striatum and cortex as expected (41). ARL13B immunodetection also detected primary cilia in the cortex and striatum but only sporadically. AC3 immunodetection was more frequent compared with ARL13B labeling providing a signal that easily distinguished primary cilia from the neuropil. Therefore, we selected AC3 as the optimal antibody to mark primary cilia on neurons in both brain regions. We also tested different species derived antibodies for each cilia marker and selected the rabbit AC3 antibody over the chicken AC3 antibody because there was lower background. In cases where the neuronal marker and the cilium antibodies were derived from the same species (anti-SATB2 and anti-DARPP-32), a sequential staining method (see Materials and Methods section and Table S1) was used as previously described (20). This staining method was also accompanied

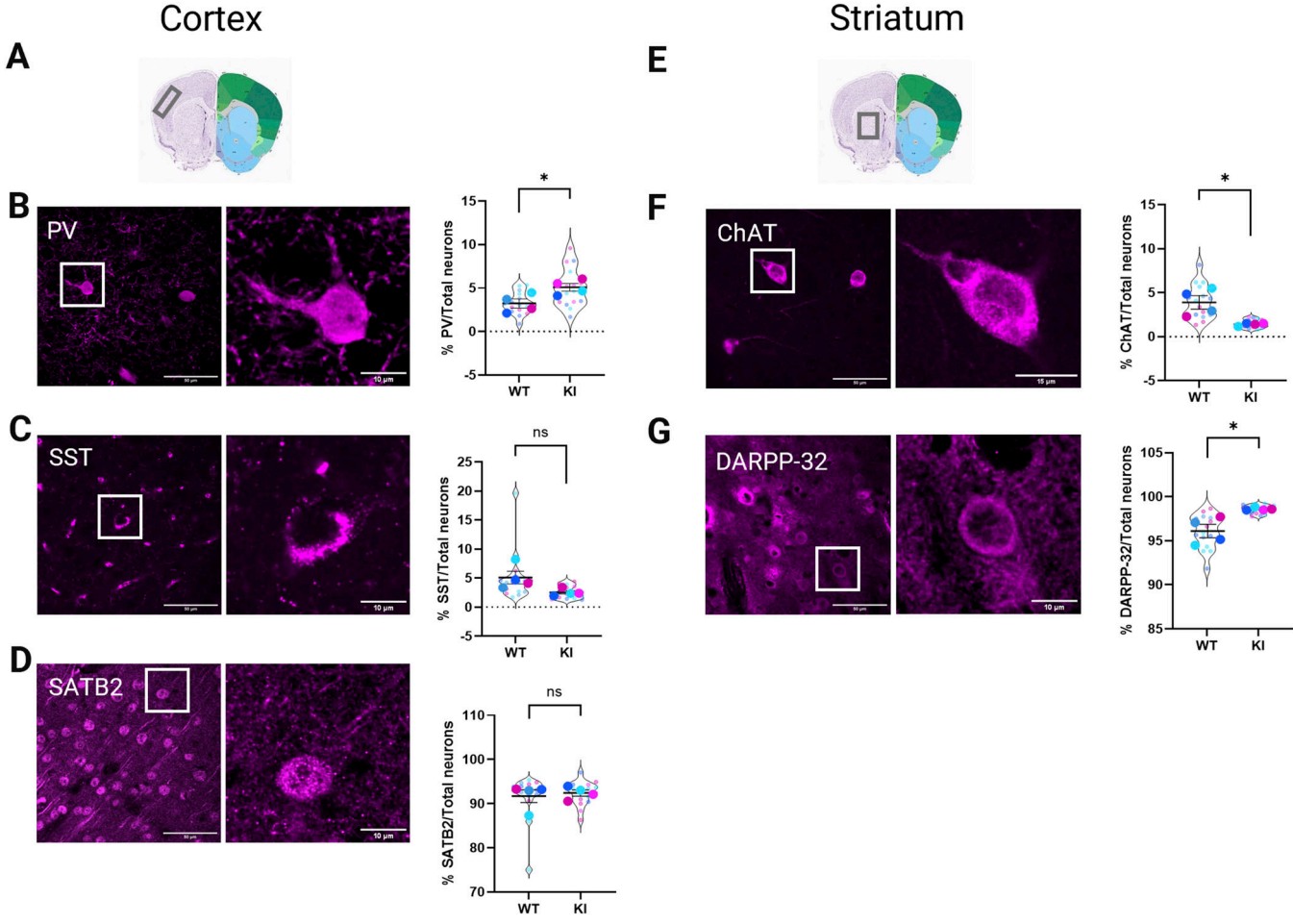

**Figure 1. Immunodetection and quantification of cortical and striatal neuronal subtypes.**
**(A)** Allen brain atlas image indicates the anatomical location of the images taken in SSCtx L4. **(B)** Representative confocal images of the SSCtx L4 showing parvalbumin (PV) immunolabeled neurons (magenta) at two magnifications and the quantification of %PV neurons to the total neurons. **(C)** Representative confocal images of the SSCtx L4 showing somatostatin (SST) immunolabeled neurons (magenta) at two magnifications and the quantification of %SST neurons to the total neurons. **(D)** Representative confocal images of the SSCtx L4 showing special AT-rich sequence-binding protein 2 (SATB2) immunolabeled neurons (magenta) at two magnifications and the quantification of %SATB2 neurons to the total neurons. **(E)** Allen brain atlas image indicates the anatomical location of the images taken in dorsal striatum. **(F)** Representative confocal images of the striatum showing cholinergic interneurons (ChAT) immunolabeled neurons (magenta) at two magnifications and the quantification of %ChAT neurons to the total neurons. **(G)** Representative confocal images of the striatum with immunolabeled dopamine and cAMP regulated phosphoprotein (DARPP-32) neurons (magenta) at two magnifications and the quantification of %DARPP-32 neurons to the total neurons. All images show Z-stack projections for each neuronal subtype marker. White boxes indicate representative neurons shown at higher magnification in the images on the right. All images are representing WT, adult male mice. Data from $GPR88^{WT/WT}$ (n = 4; 3 males, 1 female) and $GPR88^{Venus/Venus}$ (n = 4; 2 males, 2 females). Blue indicates male, pink indicates female. Violin plots show the technical replicates (separate images) for each animal as small circles and averaged data for each animal are the overlaid large circles. Statistical analysis was performed using unpaired t test, *$P$ < 0.05. Averaged data are presented as ± SEM. **(A, E)** Allen Mouse Brain Atlas, https://mouse.brain-map.org/static/atlas.

by standard controls in $GPR88^{WT/WT}$ showing that the single SATB2 or DARPP-32 antibody stain does not label structures resembling primary cilia (Fig S2A), that secondary antibodies do not label structures nonspecifically (Fig S2B) and that the antibody (Anti-GFP) used to amplify GPR88-Venus does not label subcellular structures nonspecifically (Fig S2C).

Next, brain sections containing the SSCtx L4 and the striatum from both $GPR88^{WT/WT}$ and $GPR88^{Venus/Venus}$ mice were stained by the neuronal subtype marker and the cilia marker (AC3). Primary cilia were evident on cortical neurons in confocal images from $GPR88^{WT/WT}$ mice showing inhibitory parvalbumin (Fig 2A), somatostatin (Fig 2B), and excitatory SATB2+ neurons in the SSCtx L4 (Fig 2C).

Next, we quantified the distribution of ciliated neurons in both regions and calculated the percentage of double positive cells. We quantified ciliated neurons by the following metric: the ratio of double positive cells (neuronal subtype+ and AC3+) to the neuronal subtype as a metric of cilia density within neuronal subtype. In the cortex, primary cilia were sporadically detected on inhibitory PV+ neurons for $GPR88^{WT/WT}$ (18.37% ± 3.9%) and $GPR88^{Venus/Venus}$ (17.24 ± 5.7) with no significant differences between genotypes detected (Fig 2A). The majority of the counted SST+ neurons were found to have primary cilia for $GPR88^{WT/WT}$ (63.57% ± 11.3%) and $GPR88^{Venus/Venus}$ (74.27% ± 9.8%) mice with no significant differences detected (Fig 2B). The largest population of SSCtx

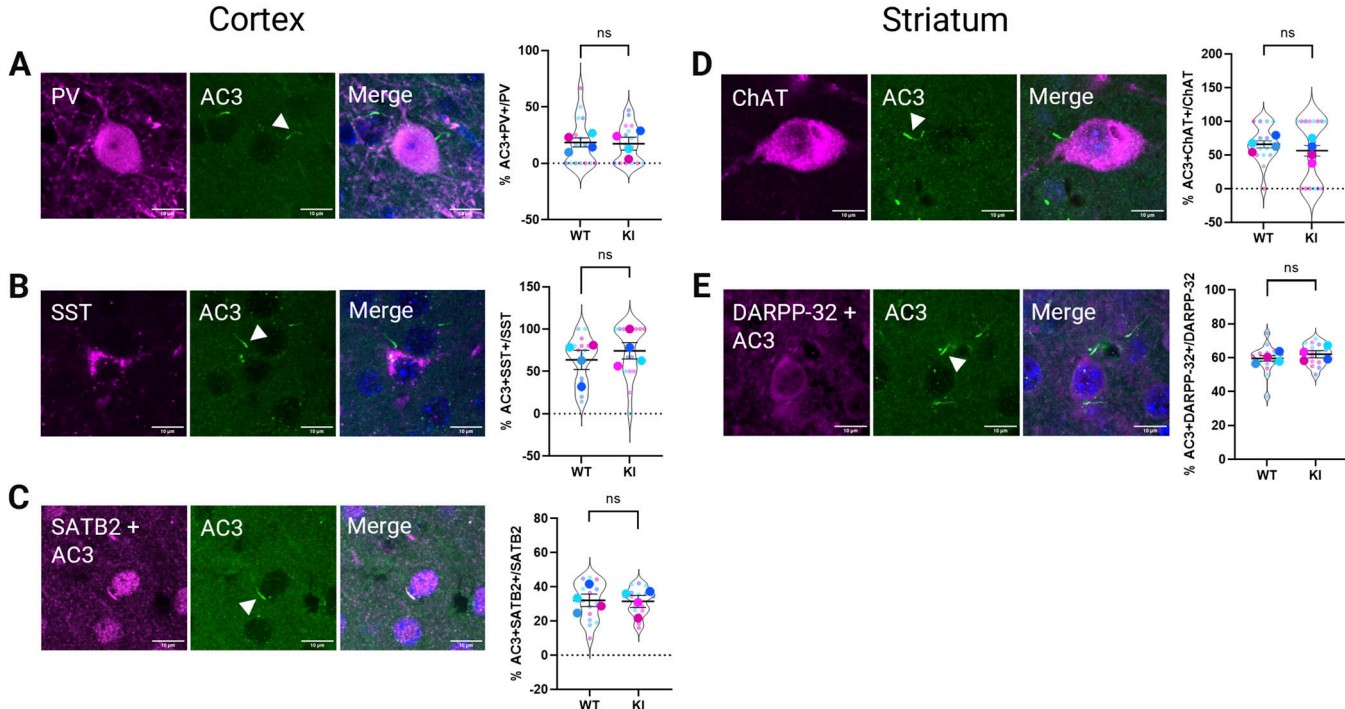

**Figure 2. Cilia densities across neuronal subtypes in the somatosensory cortex and striatum.**
Representative Z-stack projections of confocal images show immunodetection of neuronal subtype marker (magenta) and adenylyl cyclase 3 (AC3, green) labeled neuronal primary cilia with quantification of the percent double positive (subtype+/AC3+) per neuronal subtype shows the proportion of ciliated neurons for each genotype (WT = $GPR88^{WT/WT}$, KI = $GPR88^{Venus/Venus}$). **(A)** The distribution of ciliated PV inhibitory neurons in the cortex. **(B)** The distribution of ciliated SST inhibitory neurons in the cortex. **(C)** The distribution of ciliated SATB2 excitatory neurons in the cortex. **(D)** The distribution of ciliated ChAT interneurons in the striatum. **(E)** The distribution of ciliated DARPP-32 GABAergic medium spiny neurons in the striatum. Cilia localization is denoted by white arrowheads. **(C, E)** Cross-reactivity with AC3 is seen in magenta together with SATB2 or DARPP-32 as expected due to the sequential staining technique (see Materials and Methods section Table S1 and controls Fig S2). All images are representing WT, adult mice. Quantification data are from $GPR88^{WT/WT}$ (n = 4; 3 males, 1 female) and $GPR88^{Venus/Venus}$ (n = 4; 2 males, 2 females). Blue indicates male, pink indicates female. Violin plots show the technical replicates (separate images) for each animal as small circles and averaged data for each animal are the overlaid large circles. Statistical analysis was performed using unpaired *t* test and data were not significant (ns). Averaged data are presented as mean ± SEM.

L4 neurons, SATB2+, were also partially ciliated with cilia detected on $GPR88^{WT/WT}$(32.07% ± 3.6%) and $GPR88^{Venus/Venus}$ (31.43% ± 3.5%) SATB2+ neurons (Fig 2C). Collectively, this data shows that inhibitory and excitatory cortical neurons possess AC3+ primary cilia. Should GPR88 be expressed in these neuronal subtypes, we would expect the receptor to localize to primary cilia as it does in the striatum.

To determine cilia density and distribution on striatal neurons, we identified cholinergic interneurons with anti-CHAT or GABAergic MSNs with anti-DARPP-32 and anti-AC3 was used to label neuronal primary cilia. Confocal images show that primary cilia were observed on CHAT (Fig 2D) and DARPP-32 (Fig 2E) neurons. In the striatum, a high percentage of CHAT+ neurons were ciliated for $GPR88^{WT/WT}$ (65.83% ± 5.2%) and $GPR88^{Venus/Venus}$ (56.25% ± 8.1%) mice with no significant differences between genotypes detected (Fig 2D). The DARPP-32+ MSNs were also highly ciliated for $GPR88^{WT/WT}$ (59.53% ± 1.6%) and $GPR88^{Venus/Venus}$ (61.95% ± 2.1%) mice with no significant differences between genotypes detected (Fig 2E). Collectively, the distributions of ciliated cortical and striatal neuronal subtypes were similar across $GPR88^{WT/WT}$ and $GPR88^{Venus/Venus}$ genotypes suggesting that they exhibit comparable cilia densities.

## GPR88 is localized to SATB2+ excitatory neurons in the somatosensory cortex and DARPP-32+ GABAergic neurons in the striatum

Next, we asked if the observed regional differences in GPR88 localization to primary cilia are dependent on neuronal subtype. To characterize GPR88 localization to primary cilia in the adult mouse brain, we first determined which neuronal subtype expresses GPR88 using GPR88-Venus mice. We immunostained GFP to enhance fluorescence detection of Venus, alongside specific neuronal subtype markers. In the somatosensory cortex (Fig 3A), GPR88 was primarily observed on SATB2+ excitatory neurons, with no observed colocalization in inhibitory neurons (PV or SST). The quantitative data further support these findings. In the cortex (Fig 3B), GPR88 expression was only detected in SATB2+ neurons (87.39% ± 1.7%) but not in any PV or SST neurons which indicate that GPR88 is specifically expressed in excitatory neurons of the SSCtx L4. To determine if GPR88 maintains this cell-type specificity at transcript level we turned to the recently published openly available MERFISH transcriptomics dataset from the adult mouse brain (42). In agreement with our spatial proteomics, *Gpr88* transcripts were only found to co-localize in *Satb2+* neurons but not *Pvalb* or *Sst* (Fig 3C and D) in the SSCtx L4.

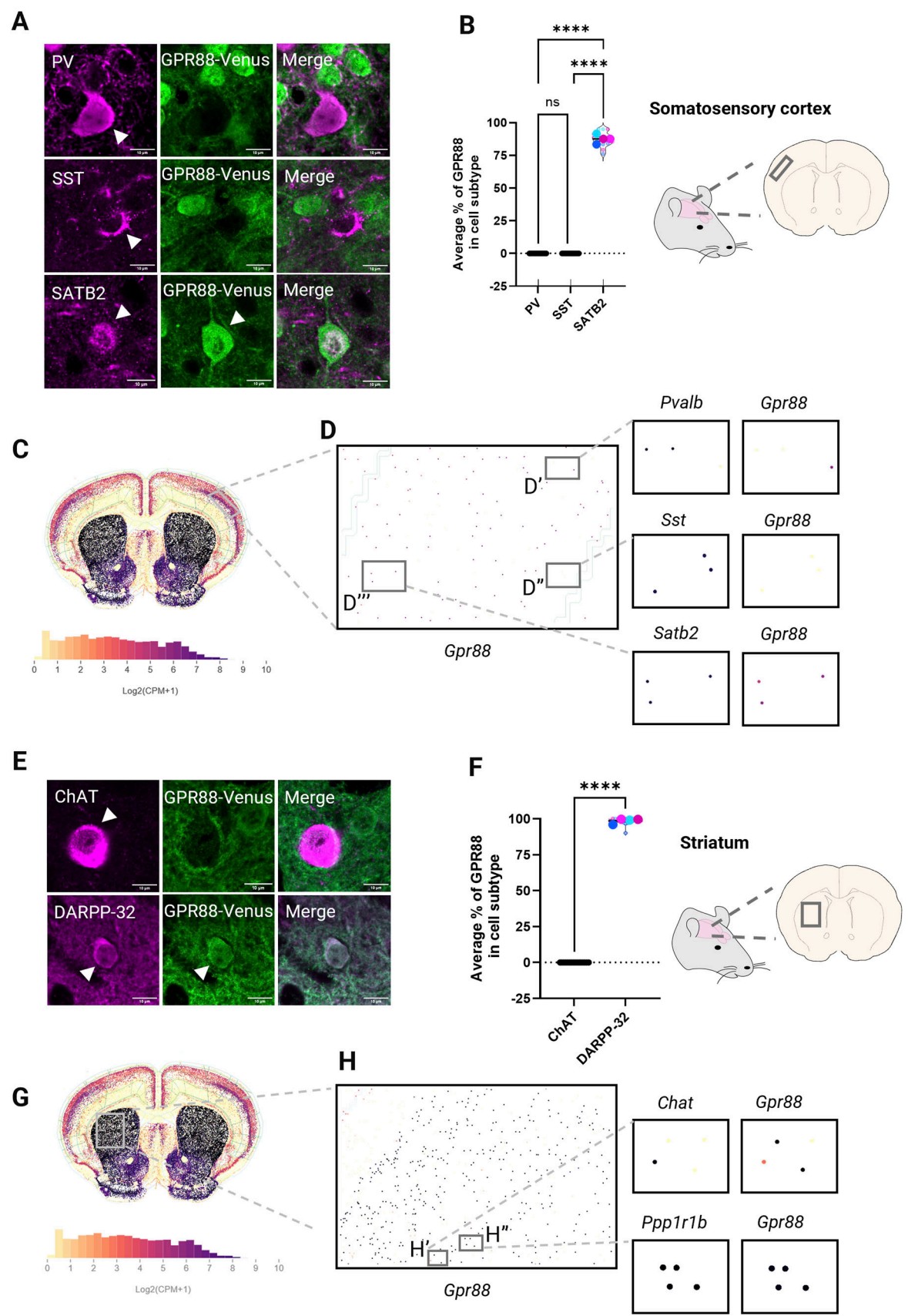

In the striatum (Fig 3E), GPR88 showed colocalization exclusively with DARPP-32+, MSNs, consistent with previous findings that GPR88 is primarily expressed in these GABAergic neurons (25, 43, 44, 45). No colocalization between GPR88 and CHAT was observed, confirming that GPR88 protein is not localized to cholinergic interneurons (44). The quantitative data in the striatum (Fig 3F) show that the average percentage of GPR88 expression in DARPP-32+ neurons was 98.62% ± 0.9% whereas CHAT+ was 0% ± 0%. This indicates that in the striatum GPR88 receptor protein localizes to MSNs. The transcriptomics (Fig 3G) were in agreement with *ChAT+* neurons containing low levels of *Gpr88* (Fig 3H') relative to the DARPP-32+ (*Ppp1r1b*) MSNs exhibiting high expression of *Gpr88* transcripts (Fig 3H"). These results highlight the selective localization of GPR88 protein and transcript to specific neuronal subtypes in both brain regions.

### GPR88 localizes to primary cilia in the striatum but not in the somatosensory cortex

To map the distribution of GPR88-Venus to primary cilia in the adult mouse brain, we examined both total and ciliary GPR88 localization in the somatosensory cortex and striatum. Confocal images revealed distinct patterns of GPR88 expression in these regions. In the somatosensory cortex, GPR88 is found on excitatory neurons that are densely clustered together (Fig 4A). Confocal images show GPR88-Venus present on soma, neuronal fibers and consistent with previous reports, GPR88-Venus colocalizes with nuclear compartments in cortical neurons (24, 44, 46, 47). However, even though AC3+ primary cilia were abundant on roughly half of the surveyed cortical neurons, no ciliary localization of GPR88-Venus was detected (Fig 4B and C). GPR88-Venus instead appeared nuclear and somatodendritic (Fig 4A). These results demonstrate that GPR88-Venus expressing cortical neurons do not target GPR88 to AC3+ neuronal primary cilia.

In the striatum, GPR88 expression is homogeneously distributed throughout the region on inhibitory MSNs. In higher magnification images, GPR88-Venus is found on dense fibers between cell bodies, somatic, and primary cilia membranes (Fig 4D). Furthermore, colocalization of GPR88-Venus with the cilia marker AC3 shows that GPR88 is present on primary cilia of mature striatal MSNs (Fig 4E). We comprehensively characterized GPR88-Venus+ neurons in the two regions. Remarkably, we did not observe any GPR88+/SATB2+ neuron with GPR88 receptors targeted to primary cilia in the cortex (Fig 4C). In

contrast, in the striatum roughly half of all double positive GPR88-Venus+/DARPP-32+ neurons target GPR88 receptors to primary cilia (Fig 4F). This is despite the fact that similar percentages of cortical neurons and striatal neurons were found to possess primary cilia (Fig 4B and F). These results from the first systematic examination of GPR88 distribution on primary cilia find that GPR88 is localized to primary cilia in a neuronal subtype specific manner, on GABAergic MSNs in the striatum but not on excitatory neurons in the somatosensory cortex.

### Cilia density and cilia length are not affected in *Gpr88-KO* mice

Since cilia are microtubule-based structures critical for axon guidance and neuronal development (48) and loss of function studies of *Gpr88* demonstrate extensive remodeling of intracortical and cortico-subcortical networks (32), we reasoned that *Gpr88* deletion might affect cilia density or length. We investigated this by quantifying cilia density and cilia length in *Gpr88-KO* and *Gpr88-WT* animals using an unbiased approach with trained artificial intelligence (AI) methods as previously reported by others (49). Brain sections from *Gpr88-WT* and *Gpr88-KO* animals containing the two regions, somatosensory cortex, and striatum, were sectioned and immunostained for AC3 to stain primary cilia and DAPI to stain nuclei (Fig 5A–D). Briefly, the confocal images were post-processed by first tracing binaries around the cilia and then training the AI to recognize cilia (see Materials and Methods section and Fig S3). In the cortex, cilia have been reported to be shorter making them less apparent as compared with the striatum (41) where cilia are longer and easier to visualize. Indeed higher magnification confocal images show that cortical primary cilia appear shorter (Fig 5A and B) than the longer striatal cilia (Fig 5C and D). We found cortical cilia were on average significantly shorter than striatal cilia (Fig S4). Comparison of cilia densities (AC3/DAPI) revealed no significant differences between *Gpr88-WT* and *Gpr88-KO* animals in the cortex (Fig 5E) and striatum (Fig 5G). Next, we compared cilia length in the two regions. Measuring the length of thousands of primary cilia (cortex: *Gpr88-WT*, 2,653 cilia and *Gpr88-KO*, 4,842 cilia; striatum: *Gpr88-WT*, 9,400 cilia and *Gpr88-KO* 9,167 cilia) revealed no significant differences in cilia length between *Gpr88-WT* and *Gpr88-KO* animals in the cortex (Fig 5F) and striatum (Fig 5H). These results suggest that deletion of GPR88 does not affect cilia density and length in the striatum or cortex.

**Figure 3. GPR88 is localized to SATB2+ excitatory neurons in the cortex and DARPP-32+ GABAergic neurons in the striatum.**
**(A, B, C, D)** GPR88 distribution in the somatosensory cortex layer 4. **(A, B)** Representative immunofluorescence-stained brain sections that show GPR88-Venus distribution in cortical neuronal subtypes. The GPR88-Venus signal quantified as the average % of GPR88 in a cell subtype is found only in SATB2+ excitatory neurons and is not detected in inhibitory neurons (PV and SST). **(C, D)** Representative spatial transcriptomic images from the Allen Institute MERFISH dataset of adult mouse brain (Allen Brain Cell Atlas (RRID:SCR_024440) https://portal.brain-map.org/atlases-and-data/bkp/abc-atlas) shows *Gpr88* transcripts at whole brain level (C) is highest in the striatum (black). **(D)** At regional level (D) in the SSCtx L4 (gray box). *Gpr88* is absent from *Pvalb* (D') and *Sst* (D") neurons but co-localized with *Satb2* (D'''). **(E, F, G, H)** GPR88 is distributed in DARPP-32 medium spiny neurons in the striatum. **(E, F)** Confocal images of immunofluorescence-stained brain sections and quantification of the average % GPR88 in cell subtype, show GPR88 distribution in DARPP-32 but not CHAT neurons. **(G, H)** *Gpr88* at whole brain level (G, H) within the striatum. *Gpr88* transcripts are detected at high level in *Ppp1r1b* (DARPP-32) (H") medium spiny neuronss and at lower levels in *ChAT+* (H') cholinergic interneurons. **(A, E)** Confocal images representing *GPR88*$^{Venus/Venus}$ mice. **(B, F)** Violin plots show the technical replicates (separate images) for each animal as small circles and averaged data for each animal as the overlaid large circles. One-way ANOVA with Tukey's comparison test; ****P < 0.0001 and values represent the data as ± SEM. Data were obtained from *GPR88*$^{Venus/Venus}$ (n = 4) animals. Blue indicates male, pink indicates female. **(C, D, G, H)** Transcriptomics expression levels increase in intensity as expression levels increase with yellow (0, not detected) and black as maximal expression. *Gpr88* levels in the striatum are shown as a reference scale (log$_2$(CPM + 1)).

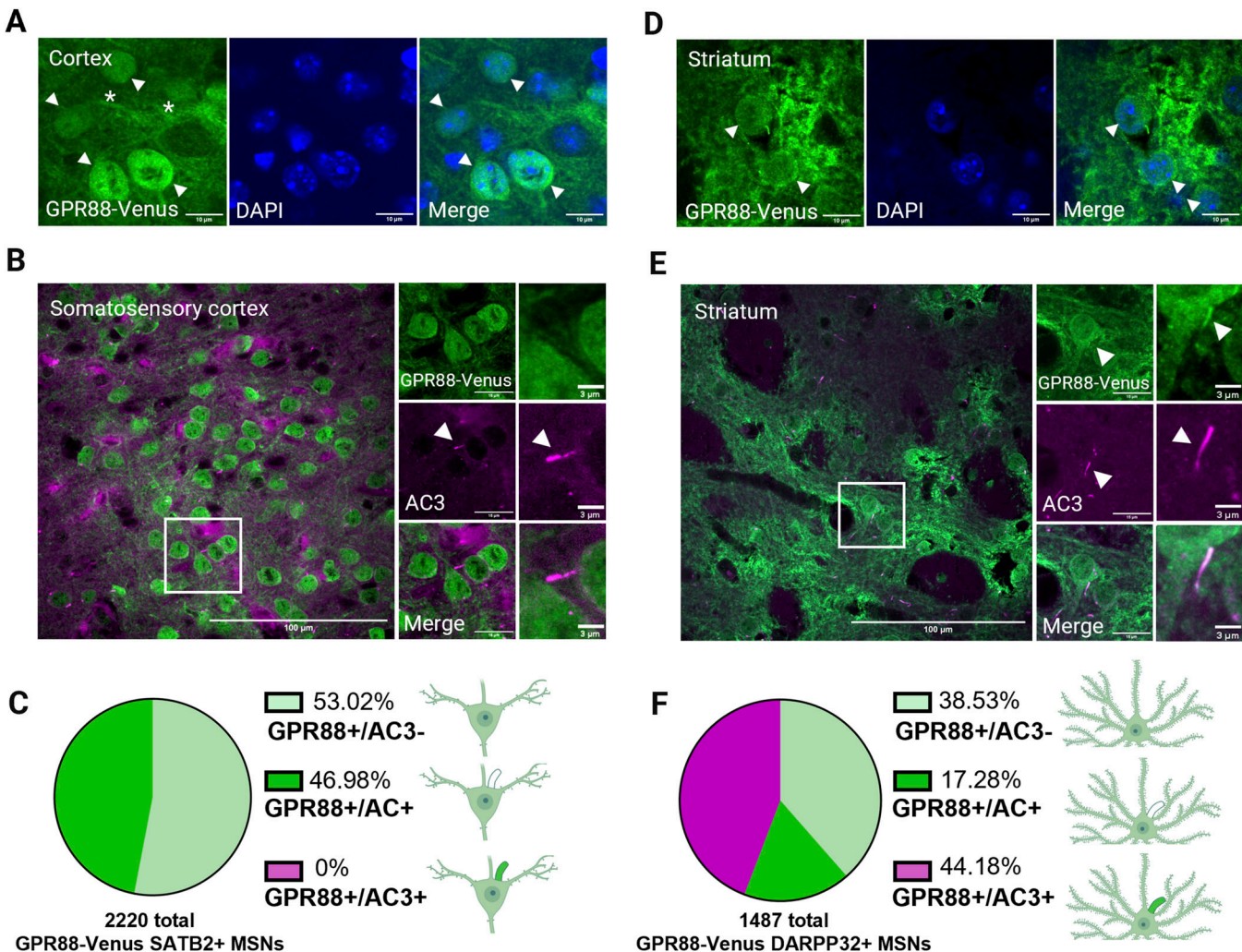

**Figure 4. GPR88 is only localized to primary cilia in the striatum, but not in the somatosensory cortex.**
**(A, D)** Colocalization of GPR88-Venus (green) and DAPI (blue) in cortical and striatal regions. In merged images, arrowheads indicate GPR88-Venus cell bodies. Asterisks indicate neuronal fibers. **(A)** In the cortex, GPR88-Venus colocalizes with DAPI in the nucleus. **(D)** In the striatum, GPR88-Venus is localized to the plasma membrane and primary cilia. **(B)** Confocal images show the overall expression of the GPR88 receptor in SSCtx L4. GPR88-Venus fibers are present, but no ciliary signal is detected; there is no colocalization with AC3. **(C)** The pie chart shows the total distribution of GPR88-Venus+ neurons in SSCtx L4. GPR88-Venus+ neurons were categorized as depicted in schemes as either not ciliated (GPR88+/AC3−), ciliated but lacking GPR88 in primary cilia (GPR88+/AC3+) or ciliated and localizing GPR88 to cilia (GPR88+/AC3+). **(E)** Confocal images show the overall expression of GPR88 in the striatum. Higher magnification images show GPR88 with the ciliary marker AC3, indicating colocalization to primary cilia. **(F)** The pie chart shows the number of GPR88-Venus+ neurons detected in the striatum that are not ciliated (GPR88+/AC3−), are ciliated but lack GPR88 in primary cilia (GPR88+/AC3+somatodendritic), or are ciliated and localize GPR88 to cilia (GPR88+/AC3+somatodendritic+primary cilia). **(B, E)** Data and images were obtained from $GPR88^{Venus/Venus}$ (n = 4) animals.

## Discussion

GPR88 is an orphan receptor highly expressed in the striatum where it modulates dopamine and thus is a potential target for neuropsychiatric disorders such as ADHD, substance abuse, and Parkinson's disease (21, 45, 50, 51, 52). Recent advancements in single cell transcriptomics reveal that neurons express dozens of GPCRs rather than a single GPCR (53). These findings suggest that determining the spatial organization of receptors is integral to understanding GPCR biology (53, 54). To improve drug targeting strategies for GPR88, a better understanding of its spatial organization is needed. Here, we show that GPR88 localization to

primary cilia is specific occurring in a cell-type and regional manner. Specifically, we show that GPR88 (1) localizes to primary cilia on inhibitory DARPP-32+ GABAergic MSNs in the striatum; (2) is excluded from primary cilia on GPR88+ excitatory SATB2+ neurons in the cortex; and (3) deletion does not impact cilia density or length in the cortex or the striatum. This work establishes neuronal subtype specific regulation of GPR88 localization to primary cilia. Ciliary targeting of GPCRs is well known to be receptor-selective, meaning that only specific GPCRs are able to cross the diffusion barrier to localize to cilia while others are excluded. The present results expand our understanding of this selective process by providing a clear example of cell type-specificity in GPCR targeting

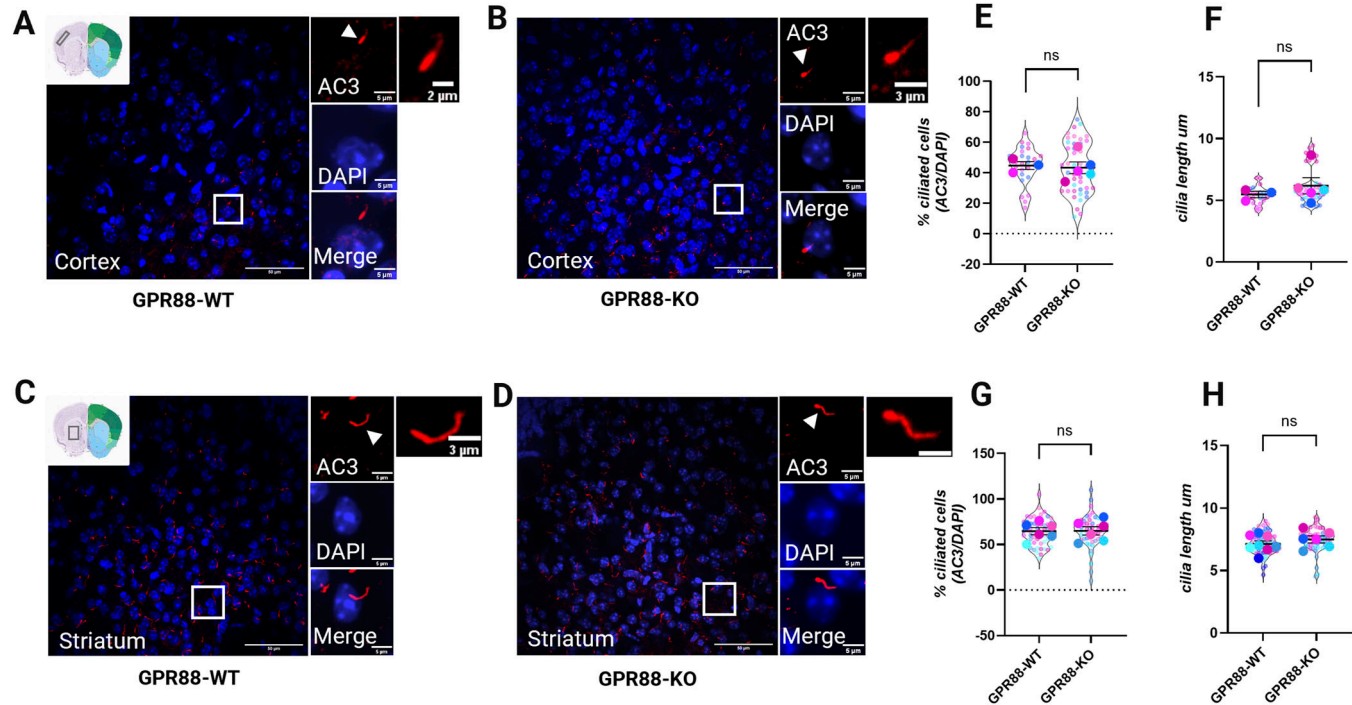

**Figure 5. Cilia length and density are unaffected by *Gpr88* depletion.**
**(A, B, C, D)** Representative confocal images showing AC3 labeled primary cilia distribution in (A, C) *Gpr88-WT* and (B, D) *Gpr88-KO* mice. **(E)** Bar graph displays the percentage of ciliated cells in the cortex by genotypes, which is measured by AC3/DAPI. **(F)** Graph shows the average cilia length in *µ*m for both genotypes in the cortex. **(G)** Bar graph shows the percentage of ciliated cells in the striatum measured by AC3/DAPI. **(H)** Average of cilia length is shown in *µ*m in the striatum. Violin plots show the technical replicates (separate images) for each animal as small circles and averaged data for each animal are the overlaid large circles. Statistical analysis was performed using unpaired *t* test and data were not significant (ns). Averaged data are presented as mean ± SEM. *Gpr88-WT* n = 3 (cortex), n = 6 (striatum), *Gpr88-KO* mice n = 5 (cortex), n = 6 (striatum). Blue indicates male, pink indicates female.

to cilia. To our knowledge, these results are the first to show cell type-specific ciliary targeting of any GPCR in vivo. Such cell type-specificity adds to a growing body of evidence supporting the general hypothesis that the primary cilium is functionally important for neuromodulation, and it opens the door to future studies testing this hypothesis using cell-type specific tools to alter GPR88 localization.

### GPR88 localizes to excitatory SATB2+ spiny stellate cells in the cortex

Our investigation into the hypothesis that GPR88 localization to primary cilia might be regulated by neuronal subtypes yielded clear findings at cellular and subcellular levels. At the cellular level, we found that GPR88 is restricted to excitatory SATB2+ spiny stellate cells. These cells are primarily found in the SSCtx L4 and receive input from the thalamus participating in cortical functions including sensory processing and information transfer between cortical areas (55, 56). A functional implication for these neurons in sensorimotor connectivity is supported by our previous results from brain-wide fMRI, where loss of GPR88 was shown to have significant alterations in sensorimotor cortical connectivity (32) and from our behavioral investigation of sensory processing deficits in *Gpr88-KO* animals which exhibit reduced tactile responses (24). In contrast with another observation (46), GPR88 was not observed on any of the inhibitory cortical

interneurons we examined, including those expressing PV and SST. This discrepancy may arise from study differences either in the region of focus in the cortex or GPR88 detection methods. Notably, the MERFISH transcript dataset in the Allen Brain Cell Atlas also supports our finding that *Gpr88* localizes specifically to SATB2+ neurons in the SSCtx L4 (Fig 3). Future studies that specifically manipulate GPR88/SATB2 cortical neurons will clarify the role for GPR88 neuromodulation in sensory processing.

### GPR88 localizes to DARPP-32 MSNs in the striatum

The striatum is a region known for high expression of GPR88 (43, 44, 45), particularly in MSNs. However, the functional role of GPR88 at the primary cilium of MSNs remains poorly understood. Our results show that GPR88 is localized specifically to DARPP-32+ MSNs and not to cholinergic interneurons (CHAT+) (Fig 3C) as previously reported by others (44). These findings support a functional role for GPR88 MSNs in mediating direct and indirect dopamine pathways and behavior in the striatum as supported extensively by *Gpr88-KO* studies (25, 50, 57, 58, 59, 60).

### Absence of GPR88 in cortical primary cilia

At the subcellular level, we found that GPR88 localizes to the cytoplasm and nucleoplasm as originally reported by reference 46

but is excluded from the primary cilium on cortical SATB2+ neurons. Our finding of GPR88 exclusion from primary cilia in the cortex has not been mechanistically determined and could arise from several factors. GPR88 localization to cilia may be regulated by developmental control as it was shown to change levels of enrichment at cilia in the striatum as the animal matures (24). Future studies using GPR88-Venus animals to map GPR88 expansion to cilia across developmental time points in the cortex may help reveal temporal cilia dynamics of GPR88 in the cortex. In addition to, distinct molecular weight species of GPR88 could account for our findings; however, similar full-length GPR88 forms have been detected in cortical and striatal lysates (47). Cis-acting ciliary targeting sequences could also be a factor and are generally found on the receptor carboxyl terminus (C-terminus) (61), the same receptor region that receives post-translational modifications (PTM) that impact localization (62). It is conceivable then that GPR88 might require a PTM on the C-terminus to be targeted to the primary cilium in cortical neurons. Notably, GPR88 localization to the cilium has not been detected anywhere in the brain by a C-terminus recognizing antibody detection method (44, 46) suggesting it does not recognize ciliary GPR88. Recently, trans-acting factors have also been identified in transport to cilia. For example, MC4R, was shown to require an accessory protein, MRAP2, to localize to primary cilia in IMCD3 cells (63) and Rhodopsin has long been shown to require a set of specific accessory proteins to localize to the cilia-like outer segment (64). In non-neuronal cells, GPR88 transport to the primary cilium has been shown to require a ciliary adaptor protein, TULP3 (65) and it is tempting to speculate that TULP3 or a yet to be identified ciliary adaptor protein may be specifically lacking in cortical neurons. Future work is needed to identify the cis and trans-acting mechanisms of GPR88 delivery to primary cilia. Collectively, these findings suggest GPR88 localization to primary cilia may be regulated by both temporal and cell-type specific mechanisms.

### GPR88 localization to primary cilia on MSNs

In addition to, we find that GPR88 is localized exclusively to primary cilia in the striatum and not the cortex (24). Plasma membrane localization of GPR88 in the striatum, along with its presence on primary cilia, provides a novel perspective on the potential functional significance of GPR88 in MSNs, particularly in cAMP signaling. The primary cilium is known to be a restricted compartment that amplifies cAMP signaling (66). The enrichment of GPR88 at the primary cilium suggests that the Gαi-GPCR inhibits ciliary cAMP signaling in MSNs. In the paraventricular nucleus of the hypothalamus, overexpression of a constitutively active GPR88 was shown to decrease the activity of ciliary localized Gαs-coupled MC4R (14). It is interesting to speculate that GPR88 could play a similar role in MSNs, perhaps insulating and buffering the cAMP signal generated by dopamine receptors as has been shown for D1 and GPR88 in IMCD3 cells (18, 23). GPCR localization to primary cilia is generally thought to be dynamic (67). Our study also sheds light on the dynamic role for GPR88. In the striatum, it is our unpublished observation that GPR88 is not uniformly distributed along the cilium, with a subset of cilia displaying accumulation of

GPR88 at one end of the cilium. This partial localization of GPR88 within cilia suggests that the receptor may be involved in localized signaling mechanisms at the ciliary base or tip in MSNs. Alternatively, it may suggest that GPR88 is dynamically trafficked in and out of the cilium depending on signaling similar to smoothened which depends on Hedgehog signaling (4, 68, 69). In addition, since DARPP-32 labels all MSNs it would be worthwhile for future studies to determine whether ciliary GPR88 is preferentially localized to D1 or D2 MSNs, or whether it is present on cilia in both populations.

Recent studies in striatal neurons and other cell lines have demonstrated that the local receptor environment is critical for determining the cAMP/PKA response of a GPCR (70, 71). The local receptor environment at the primary cilium through modulating cAMP/PKA signaling likely affects neuromodulation either through transcriptional regulation (66) or regulation of synaptic neurotransmission (6, 7, 11, 72 *Preprint*) regulating complex biological processes such as energy homeostasis (19) or cognition (73). Future work precisely manipulating GPR88 localization at MSN primary cilia and measuring signaling effects will clarify the role ciliary GPR88 localization plays in dopamine regulated behavior. Collectively, these results suggest that GPR88 is selectively targeted to primary cilia in the adult striatum and its dynamic localization points to a potential role for ciliary GPR88 signaling in the regulation of striatal neuronal function in the mature brain, particularly in the context of neuropsychiatric disorders that involve altered dopaminergic signaling.

### Limitations of the study

The results in this study were derived from data collected from small cohorts of animals (n = 3–6 per group). While this sample size has been typically used to describe neuroanatomy in the neuroscience field these results may not reflect the expression across the entire cortex and striatum. Our study can therefore reach the limited conclusion that GPR88 does not localize to primary cilia in the thousands of neurons that we sampled but we cannot rule out that a more comprehensive method such as stereology might obtain different findings. We find the percentage of ciliation across neuronal subtypes to be on the lower side as compared with advanced electron microscopy methods (6, 7) which generally identify primary cilia on most neurons. This discrepancy is likely due to the technically challenging assignment of cilia to soma in post-fix brain sections. We only assigned cilia to cell bodies when the cilium could clearly be traced to the soma. Consequently, many AC3+ cilia were detected but not assigned to a specific neuronal subtype if they did not meet this criterion.

# Materials and Methods

### Animals

GPR88-Venus knock-in mice were generated at the Institut Clinique de la Souris at Illkirch-Graffenstaden, France. Only mature mice were used aged 8–18 wk, male and female were bred in-house at

the University of California, San Francisco and cared for by Laboratory Animal Resource Center staff. Mice were grouped in cages with free access to food and water and handled according to the guidelines set by the facility. All the procedures were performed according to the guidelines of the Canadian Council of Animal Care and were approved by the Institutional Animal Care and Use Committee (IACUC) at the University of California, San Francisco. Mice were housed with littermates and maintained on a 12-h light/dark cycle with access to food and water.

*Gpr88-KO* mouse brains were extracted at the McGill University/Douglas Hospital Research Institute, Montreal Canada and histology procedures were performed at the University of California, San Francisco. *Gpr88-KO* mice were generated as previously described (25).

## Brain tissue preparation

Mice were given $CO_2$ for anesthesia. WT and knock-in, female and male mice were intracardially perfused with 10 ml of PBS (cat# 14040216; Life Technologies), followed by 50 ml of 4% PFA/PBS solution (cat#1573590; Thermo Fisher Scientific). Heads were cut and brains were extracted into 10 ml 4% PFA and PBS and refrigerated overnight. The whole brains were sunk and were moved into 30% sucrose in PBS for cryoprotection for 48 h at 4°C. Brains were mounted on deep base molds in dry ice using O.CT. compound (Sakura, VWR) and stored at –80°C until used in immunofluorescence. Brains were sliced in coronal sections at 30 $\mu$m on the cryostat (CM1950; Leica), sections with the regions of interest (Allen Brain Institute mouse brain atlas, images 44–65) were collected into wells with PBS and kept free floating at 4°C for immunofluorescence procedures.

## Immunohistochemistry and imaging

Sections were permeabilized with PBS-T (0.2% Triton X-100 in PBS) for 10 min and then blocked for 1 h with a blocking buffer 10% normal goat serum (cat#5425S; Cell Signaling Technology) in PBS containing 0.3% Triton X-100 (cat#T8787; Sigma-Aldrich). After, sections were incubated with primary antibodies (Table 1) in 3% normal goat serum with PBS overnight at 4°C. Tissue was then washed in PBS-T three times for 10 min. Secondary antibodies (Table 2) were diluted in the blocking buffer and incubated for 1 h at RT. Sections were then washed in PBS-T for 10 min, following a DAPI/PBS wash for 10 min, then a final 10 min PBS-T wash, slices were mounted onto slides and sealed with coverslips using ProLong gold (Thermo Fisher Scientific) and left to dry overnight. All sections were stained with chicken GFP antibody (cat#NB1001614; Novus Biologicals) to amplify the GPR88-Venus signal, DAPI (cat#D9542; Sigma-Aldrich) to stain cell nuclei and rabbit AC3 (cat#ab125093; Abcam) to label neuronal primary cilia.

For ChAT antibody immunostaining, sections were incubated with 10% normal donkey serum (cat#017-000-121; Jackson ImmunoResearch). For SATB2 and DARPP-32, sequential staining was performed as previously reported (20) (Fig 2 and Table S1). This technique involved staining for the rabbit AC3 antibody first followed by its secondary antibody. The following afternoon a separate same-species antibody was applied for the neuronal marker

followed by a secondary antibody. Using this method, the predicted false positives are to detect the cilia marker in both the cilia marker channel and the neuronal subtype marker channel but not vice versa. The GPR88-Venus signal was amplified through species distinct amplification processes and would not be expected to have any false positives. We performed control experiments in which we (1) stained the rabbit raised neuronal subtype markers (SATB2 or DARPP-32) individually from the rabbit raised cilia marker (AC3) and (2) stained only secondary antibodies and omitted the primary antibody. This approach helped us evaluate any non-specific binding of the secondary antibody (see Fig S2).

The immunofluorescence images were acquired using the CSU-W1 high speed widefield microscope (Nikon) with 60x objective lens with oil. For the striatum, four images per section were taken across two sections in the dorsal striatum to assess DARPP-32 and CHAT expression. For SATB2, parvalbumin and somatostatin, four images per section were captured in layer 4 of the somatosensory cortex. The Allen Mouse Brain was used as a reference to guide the location and regions of interest (ABI, Atlas: 44–65). Each brain region was imaged as a Z-stack with 0.26 step size.

## Image analysis and quantification

All images were manually quantified using the cell counter plugin in ImageJ. The cell counter and multi-point tools were exploited to mark cells of interest and to keep track of the number of labeled cells in each Z-stack image. Statistical analyses were performed using GraphPad Prism software. Results are presented as the mean ± SEM. Graphs were generated using GraphPad Prism.

The original analysis was performed without formal blinding of the investigator to genotype. To minimize bias, images were analyzed using predefined, objective criteria, and thresholds established before data quantification. A challenge in quantification is the potential for high background staining, which can complicate the quantification of positive neurons. To determine whether neurons were GPR88-Venus positive, imaged sections from GPR88-WT animals were stained with anti-GFP and anti-chicken 488 antibodies (Fig S2C) to measure background fluorescence levels. This background was used to set a fluorescence intensity threshold to distinguish positive signal from noise. Regions with intensity above this threshold in GPR88-Venus images were classified as GPR88-positive. During manual counting, we excluded neurons or cilia that were touching borders of the image or did not count objects that were difficult to identify. To ensure accuracy to assigning cilia, we counted a cell as AC3-positive only when the cilium could be clearly traced to the soma. These criteria were applied consistently across all images to ensure accurate and reproducible measurements.

## Length measurement and quantification of primary cilia using artificial intelligence

Measuring and assessing the structure and length of cilia can be a challenging and time-consuming task. To address this, we implemented the same AI-based analysis methods as previously reported (49) an unbiased approach of using AI to accurately measure cilia length and volume. In this method (Fig S3), AI is

**Table 1.  Primary antibodies.**

| Antibody | Host species | Dilution | Manufacturer |
|---|---|---|---|
| Anti-AC3 | Rabbit | 1:1,000 | #ab125093; Abcam |
| Anti-ChAT | Goat | 1:500 | AB144P #3426630; Millipore |
| Anti-DARPP-32 | Rabbit | 1:1,000 | #2306S; Cell Signaling Technology |
| Anti-GFP | Chicken | 1:1,000 | NB100-1614; Novus Biologicals |
| Anti-SATB2 | Rabbit | 1:2,000 | #ab34735; Abcam |
| Anti-somatostatin | Mouse | 1:500 | #14-9751-82; Thermo Fisher Scientific |
| Anti-parvalbumin | Mouse | 1:1,000 | #MAB1572; Millipore |

**Table 2.  Secondary antibodies.**

| Secondary antibody | Fluorophore | Dilution | Manufacturer |
|---|---|---|---|
| Donkey anti-chicken | Alexa 488 | 1:2,000 | A-78948; Thermo Fisher Scientific |
| Donkey anti-goat | Alexa 647 | 1:2,000 | A-21447; Thermo Fisher Scientific |
| Donkey anti-rabbit | Alexa 594 | 1:2,000 | A-21207; Thermo Fisher Scientific |
| Goat anti-chicken | Alexa 488 | 1:2,000 | A-11039; Thermo Fisher Scientific |
| Goat anti-mouse | Alexa 647 | 1:2,000 | A-21235; Thermo Fisher Scientific |
| Goat anti-rabbit | Alexa 594 | 1:2,000 | A-11012; Thermo Fisher Scientific |
| Goat anti-rabbit | Alexa 647 | 1:2,000 | A-21245; Thermo Fisher Scientific |

trained on a set of sample images to identify cilia and segment cilia on the image. Once the training is complete, the AI is applied to a new set of images where it draws a binary mask to identify cilia structures. After image segmentation, a customized general analysis (GA) is used to perform various quantitative analyses, including measurements of cilia length and intensity (49). The results are then exported into a data table, which was easily interpreted and used for further statistical analysis in Graph Pad Prism.

## Data Availability

All data underlying the research presented in the manuscript will be made available upon request.

## Supplementary Information

## Acknowledgements

We thank the *M. von* Zastrow lab for the insightful comments and feedback. We thank the UCSF Center for Advanced Light Microscopy (K Herrington and S Kim) for providing services and assistance for technical help. We thank the Julius lab and the Rubenstein lab for the use of the cryostat. We thank the Core facility ICS, Michel Bouvier and Brigitte Kieffer for the creation of the GPR88-Venus mouse. This work was supported by National Institutes of Mental Health grants R01MH12012 (M Von Zastrow and BL Kieffer), and K01MH123757 (AT Ehrlich).

## Author Contributions

YH Li Guan: data curation, formal analysis, investigation, visualization, methodology, and writing—original draft, review, and editing.
BL Kieffer: resources, funding acquisition, and writing—review and editing.
M von Zastrow: resources, funding acquisition, and writing—review and editing.
AT Ehrlich: conceptualization, resources, data curation, formal analysis, supervision, funding acquisition, validation, visualization, methodology, project administration, and writing—original draft, review, and editing.

### Conflict of Interest Statement

The authors declare that they have no conflict of interest.

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
