## [Reviewer comments · Life Science Alliance]

GPR88 localization to primary cilia in neurons is cell-type specific

Yenni Li Guan, Brigitte Kieffer, Mark von Zastrow, and Aliza Ehrlich

DOI: <https://doi.org/10.26508/lsa.202503366>

Corresponding author(s): Aliza Ehrlich, University of California, San Francisco

Review Timeline:

Submission Date:	2025-04-21
Editorial Decision:	2025-04-22
Revision Received:	2025-05-15
Editorial Decision:	2025-07-07
Revision Received:	2025-10-15
Editorial Decision:	2025-11-17
Revision Received:	2025-11-19
Accepted:	2025-11-20

Scientific Editor: Tim Fessenden

Transaction Report:

April 22, 2025

Re: Life Science Alliance manuscript #LSA-2025-03366P

Dr. Aliza Ehrlich
University of California, San Francisco
600 16th St
San Francisco 94158

Dear Dr. Ehrlich,

Thank you for submitting a presubmission inquiry related to your manuscript entitled "GPR88 localization to primary cilia in neurons is cell-type specific" to Life Science Alliance. We have considered your inquiry and we feel this work may be suitable for LSA and we invite you to submit this manuscript in full. Please note that we cannot make an editorial commitment without seeing the entire manuscript.

To upload your manuscript, please log in to your account: <https://lsa.msubmit.net/cgi-bin/main.plex>

Please don't hesitate to contact me with any questions. We are looking forward to receiving your manuscript.

Sincerely,

Tim Fessenden
Executive Editor
Life Science Alliance

July 7, 2025

Re: Life Science Alliance manuscript #LSA-2025-03366

Dr. Aliza Ehrlich
University of California, San Francisco
600 16th St
San Francisco 94158

Dear Dr. Ehrlich,

Thank you for submitting your manuscript entitled "GPR88 localization to primary cilia in neurons is cell-type specific" to Life Science Alliance. The manuscript was assessed by expert reviewers, whose comments are appended to this letter.

Reviewers were mixed in their overall enthusiasm for these findings. Reviewers 1 and 3 remarked on the importance of these observations towards understanding the neuronal populations and localization of this GPCR as well as its requirement for neuronal cilia. Specific points raised by all reviewers relate to methodology, controls, and validation needed to better support the central findings. A suitable revision must address all points by Reviewers 1 and 3, in particular point 1 by Reviewer 1 on sample size, which was also noted by Reviewer 2. A revised manuscript must also validate key observations using a different antibody (such as CFAP300) and use KO samples to confirm specificity of the GPR88 antibody as requested by Reviewer 2.

Thank you for this interesting contribution to Life Science Alliance. We are looking forward to receiving your revised manuscript.

Sincerely,

B. MANUSCRIPT ORGANIZATION AND FORMATTING:

Reviewer #1 (Comments to the Authors (Required)):

The manuscript titled "GPR88 localization to primary cilia in neurons is cell-type specific" investigates the cell-type specific localization of the G protein-coupled receptor (GPCR) GPR88 to primary cilia in neurons of the brain, with a focus on understanding the receptors spatial distribution in the cortex and striatum.

GPR88 has been observed in primary cilia in overexpression systems and native neurons previously. This manuscript seeks to: (1) determine whether GPR88 localizes to primary cilia in a cell-type specific manner in the brain; and (2) assess whether GPR88 influences neuronal cilia structure. The experimental also analyze neuronal subtype distribution in the cortex and striatum in the GPR88-Venus mouse model.

The manuscript makes several findings that build from previous reports of differential ciliary localization of GPR88 in the cortex and striatum. First, they report that GPR88 localizes to both the somatodendritic compartment and primary cilia of DARPP-32+ GABAergic medium spiny neurons (MSNs) in the striatum. In contrast, GPR88 is found in the somatodendritic and nuclear compartments of SATB2+ excitatory neurons in the somatosensory cortex and is not evident within the primary cilia. The novelty in this work resides in the delineation of the specific neuronal sub-types expressing GPR88 in the cilia. The manuscript also determines that cilia density or length in either the cortex or striatum are unaffected by GPR88. Finally, the manuscript illustrates variable distribution of GPR88 within striatal cilia-either concentrated at one end or distributed along the entire cilium. Overall, the manuscript is well written. The methods are detailed and explicit. The description of GPR88 ciliary localization in individual neuronal subtypes is novel. There are several concerns regarding the manuscript as well.

(1) Figures 1, 2, and 3 appear to minimize any potential differences in the less abundant neuron populations (example: PV, SST and CHAT) between genotypes. The Y scale of the graphs focus on abundant neuron populations, causing details about less numerous neurons to be lost. Further, the statistics illustrate relative abundance of one neuronal subtype versus another within the same genotype and not comparing between the genotypes. Thus, there is a disconnect between the experimental rationale/text and the data. Admittedly, the authors state that "No significant differences were observed..." Yet, whether these studies have included enough replicates is uncertain. This concern is emphasized in the description of Figure 2E (Line 288). The authors indicate that there is a significant difference between genotypes for DARPP-32 positive cells within the text, but this difference is not indicated on the graph (2E). Surprisingly, the authors state that this result is due to "technical factors" or "counting considerations". The authors have previously shown that there are developmental alterations in the brains of GPR88 Venus/Venus mice-so changes in cell populations are possible. Perhaps small sample size is an issue in identifying the effect of GPR88?

(2) The authors show that GPR88 disruption does not affect cilia length and state that this indicates that GPR88 does not play a role in assembly or formation. Cilia length can be regulated by ciliary protein expression and signaling. Perhaps it would be better to state that GPR88 does not affect cilia "morphology" rather than "assembly/formation".

(3) Supplemental Figure 1: AC3 chicken and Arl13b rabbit antibodies do not appear to robustly identify cilia (Figure S1) and do not mimic the AC3 rabbit data. Could the importance of this observation be clarified? Is this graph provided to lend insight into the effect of available antibodies?

(4) The statement on line 278 "Collectively, this data shows that all cortical neurons possess AC3+ primary cilia..." implies that the data show that 100% of all neurons have AC3+ primary cilia. Perhaps this should read "Collectively, this data shows that all subtypes of cortical neuron possess AC3+ primary cilia..."?

(5) In Line 393, the authors state "This pattern of partial GPR88 localization was observed in approximately 40% of the GPR88-Venus brain sections". Should this be 40% of total cilia or 40% of cells? If the differential distribution of GPR88 to one end of the cilia versus another is dependent on the section, it might suggest that differential localization is affected by the fixation/antibody staining process.

(6) Some of the data appear to have very low "n" values (3 for Figure 5G and H). Is this 'n' statistically appropriate given the variability between individual samples?

Reviewer #2 (Comments to the Authors (Required)):

This manuscript documents the ciliation status and localization of GPR88 in various mouse brain regions. They previously reported that GPR88 widely localizes to ARL13B labeled primary cilia in the striatum whereas cortical neurons do not localize GPR88 to primary cilia. Here they categorize this further. Unfortunately, significant additional work would be needed before publication can be recommended. With the additional data, the findings would be of limited impact as they are descriptive and do not teach us much more than GPR88 does not influence cilia length.

Comments:

The power of the analysis is not adequate. "Four images per section and 2 sections" led to the reported conclusions. The striatum represents ~100 X 30µm thickness sections. Stereology would involve at least 20 sections analyzing every fifth section. The somatosensory cortex is a perfectly ciliated brain region. Thus the findings are yet not adequately supported.

Their RNA analysis in Ref. 24 shows almost no expression in somatosensory cortex, despite venus expression. How do the authors explain this?

It is well established that AC3 is a neuronal cilia marker while Arl13B is better at detecting astrocyte cilia. Recently, CFAP300 was reported to be the best pan-cellular cilia marker. doi: <https://doi.org/10.1101/2024.10.20.619273>. Any absence of cilia should be validated with another pan-cellular cilia marker such as CFAP300 to be sure that lack of staining does not reflect loss of ciliary marker rather than loss of cilia.

Line 267. "The ratio of double positive cells (neuronal subtype+ and AC3+) to the cilia marker, AC3, as a metric of overall density across all ciliated cell subtypes or to the neuronal subtype as a metric of cilia density within neuronal subtype" This is not helpful. Just state the ciliation percent for each neuronal type.

Line 304. Dropviz.org gives all expression across striatum and shows no transcript in ChAT neurons.

The authors need to show the GPR88 staining on sections of their GPR88 knockout mouse to validate their antibody.

"Nuclear" localization of a 7pass transmembrane protein merely reflects new protein translation in the ER which may vary across cell types. The cited reference for GPR88 binding chromatin associated proteins should be omitted as the work is of poor quality. Again, this is from a region that has very little endogenous GPR88 expression? How do you explain the staining?

For the striatum, the cilia have been studied and quantified extensively and are not referenced. Please state precise age of mice as ciliation status and length varies with age. Also, all cells showed some soma staining as would be expected for a GPCR.

Reviewer #3 (Comments to the Authors (Required)):

In this study by Guan et al., the localization of an orphan G-protein coupled receptor (GPCR), GPR88, is characterized in two brain regions. They found that depending on the cell type, GPR88 can be enriched in the primary cilia, the soma, and/or the nucleus. They also found that GPR88 knockout does not impact ciliogenesis or cilium length. Overall, this study is important as it provides more information on an orphan receptor that is of interest as a target for addiction. However, the study heavily relies on immunofluorescent imaging and analysis, and the methodology appears to be subjective and not clearly explained. To improve the robustness and replicability of this study, the images should be re-analyzed in a more objective fashion, with the details explained in the methods sections.

Major Comments

- This paper relies heavily on analysis of immunofluorescent images, however in most cases the images were analyzed subjectively by an investigator in ImageJ, with no blinding. For example, how did the researcher determine what was GPR88+ in Figure 1? One would consider staining the GPR88 WT sections with the anti-GFP and anti-chicken 488 antibodies and measuring fluorescence to obtain a background level of fluorescent intensity. Then, in the GPR88-Venus images one can segment based on regions that are above this threshold. This sort of thresholding must be done for image analysis in the paper where you must identify an object as 'positive' for a particular antibody staining. If this was done, it is not clearly explained in the current manuscript.

- Line 393: The authors claim a concentrated GPR88 localization was found in 40% of the GPR88-Venus brain section. However, they do not put this quantification into the figure. How was this quantification performed? How was it objectively determined when a cilium had a uniform versus concentrated distribution of GPR88? This analysis, as it is explained currently, seems subjective. An automated image analysis pipeline is needed to determine the distribution of GPR88 along the length of the cilium.

- Line 184: Just saying one used an 'AI' is not sufficient explanation of the experimental methods. Did the authors use the same

software that the researchers used in reference #27? Were there any modifications to the protocol? Please provide more detail so that another researcher could replicate your methods.

Minor Comments

- Line 26: It would be clearer to say "In contrast, in the somatosensory cortex, GPR88 localizes to somatodendritic and nuclear compartments and not the primary cilia of excitatory spiny stellate neurons"
- Line 197: Why were these two regions chosen for analysis? Are they of particular interest for the function of GPR88. More explanation is warranted for this choice.
- Line 269: Based on this analysis, the authors only identified cell type for about 50% of the AC3+ cilia that were identified. Could the authors speculate in the discussion which cell types they did not stain for which could account for the other 50% of cilia?
- Line 273: Based on EM data (DOI: 10.1016/j.cub.2024.04.043), all neurons in the mouse cortex, at least in the visual cortex, are ciliated. Please include a discussion about why the data show only 68-17% ciliation in this study.
- Line 278: These data do not show that all cortical neurons possess AC3+ cilia. In fact, the authors are showing that for some neuron subtypes, only 20% possess AC3+ cilia. Please rephrase.
- Line 499: By mutationally active do you mean constitutively active? At least, that paper refers to it as "constitutively active" and I believe that is the correct nomenclature.
- Figure 4F: Do the cells which have GPR88 in the cilium also have it in the somatodendritic compartment? If so, the pie chart should be labeled GPR88+/AC3+ primary cilia and somatodendritic.
- Figure 5: The coloring on this bar chart is different than all the other bar charts in the paper. It would be better to make this consistent.

LSA-2025-03366

GPR88 localization to primary cilia in neurons is cell-type specific

Dear Referees,

We appreciate that the 3 reviewers and editorial staff at Life Science Alliance are willing to consider our manuscript for publication, and we hope they find our revised manuscript satisfactory. We thank the reviewers for their constructive comments, suggestions and review of our manuscript. To address the important issues raised by the reviewers, we have conducted additional experiments, improved the analyses, reanalyzed data to display the data better and revised the manuscript according to the reviewer's suggestions. We believe that the new results and contents greatly improve the quality of the manuscript.

Please find a point-by-point response to each comment below.

With my best regards,

Aliza Ehrlich

The original comments of each reviewer are in black and proposed responses (comments, new experiments and analyses) in blue.

Reviewer #1:

(1) Figures 1, 2, and 3 appear to minimize any potential differences in the less abundant neuron populations (example: PV, SST and CHAT) between genotypes. The Y scale of the graphs focus on abundant neuron populations, causing details about less numerous neurons to be lost. Further, the statistics illustrate relative abundance of one neuronal subtype versus another within the same genotype and not comparing between the genotypes. Thus, there is a disconnect between the experimental rationale/text and the data. Admittedly, the authors state that "No significant differences were observed..." Yet, whether these studies have included enough replicates is uncertain. This concern is emphasized in the description of Figure 2E (Line 288). The authors indicate that there is a significant difference between genotypes for DARPP-32 positive cells within the text, but this difference is not indicated on the graph (2E). Surprisingly, the authors state that this result is due to "technical factors" or "counting considerations". The authors have previously shown that there are developmental alterations in the brains of GPR88 Venus/Venus mice-so changes in cell populations are possible. Perhaps small sample size is an issue in identifying the effect of GPR88?

RESPONSE: We thank the Reviewer for this thoughtful and constructive feedback. In response, we have reanalyzed the data and revised the figures to more clearly display all collected datapoints. Throughout the revised manuscript, data are now presented as violin plots, with individual animal values shown as small circles and the mean \$\pm\$ SEM indicated by larger overlaid circles. We have also reorganized the data presentation so that each neuronal population is displayed and compared separately, enabling clearer visualization of differences between genotypes. The corresponding text has been updated to reflect these changes, and speculative interpretations have been removed. Finally, we have added a paragraph in the *Discussion* (titled "Limitations of the Study") to explicitly acknowledge the constraints of small sample sizes and the implications for interpreting these findings.

(2) The authors show that GPR88 disruption does not affect cilia length and state that this indicates that GPR88 does not play a role in assembly or formation. Cilia length can be regulated by ciliary protein expression and signaling. Perhaps it would be better to state that GPR88 does not affect cilia "morphology" rather than "assembly/formation".

RESPONSE: We thank the Reviewer for this insightful suggestion. We have revised the statement to clarify that our findings indicate no effect of GPR88 disruption on cilia morphology, while acknowledging that potential roles in cilia assembly or formation cannot be excluded.

(3) Supplemental Figure 1: AC3 chicken and Arl13b rabbit antibodies do not appear to robustly identify cilia (Figure S1) and do not mimic the AC3 rabbit data. Could the importance of this observation be clarified? Is this graph provided to lend insight into the effect of available antibodies?

RESPONSE: We thank the Reviewer for this helpful feedback. In the revised manuscript, we have updated the images in Supplemental Figure 1 to higher magnification to facilitate clearer visualization and comparison of antigen labeling across conditions. We have also clarified in the text that the purpose of testing different cilia-labeling antibodies was to identify a neuronal cilia marker with broader neuronal distribution, as ARL13B predominantly labels glial cilia in the mature brain. In addition, we have elaborated on our rationale for testing antibodies raised in different species.

(4) The statement on line 278 "Collectively, this data shows that all cortical neurons possess AC3+ primary cilia..." implies that the data show that 100% of all neurons have AC3+ primary cilia. Perhaps this should read "Collectively, this data shows that all subtypes of cortical neuron possess AC3+ primary cilia..."?

RESPONSE: We thank the Reviewer for this helpful clarification. The statement has been revised to read: "Collectively, this data shows that inhibitory and excitatory cortical neurons possess AC3+ primary cilia."

(5) In Line 393, the authors state "This pattern of partial GPR88 localization was observed in approximately 40% of the GPR88-Venus brain sections". Should this be 40% of total cilia or 40% of cells? If the differential distribution of GPR88 to one end of the cilia versus another is dependent on the section, it might suggest that differential localization is affected by the fixation/antibody staining process.

RESPONSE: We thank the Reviewer for this helpful comment. In response, we have removed the corresponding data and figure from the revised manuscript. The observation is now mentioned briefly as an unpublished finding in the *Discussion* section.

(6) Some of the data appear to have very low "n" values (3 for Figure 5G and H). Is this 'n' statistically appropriate given the variability between individual samples?

RESPONSE: We thank the Reviewer for this thoughtful comment regarding sample size. We acknowledge that an n of 3 animals per genotype is relatively small; however, this number is consistent with common standards in the field for studies involving genetic comparisons. Importantly, each animal contributed data from hundreds of individual cilia measurements, providing a large number of technical replicates that yield robust and precise estimates within each biological replicate. While we recognize the limitations inherent to small n values, the extensive sampling at the technical

level increases the reliability and statistical power of our comparisons. In addition, all data are now presented as violin plots to display the full distribution of collected measurements, and we have added a paragraph in the *Discussion* (titled "Limitations of the Study") explicitly acknowledging these considerations.

Reviewer #2:

The power of the analysis is not adequate. "Four images per section and 2 sections" led to the reported conclusions. The striatum represents ~100 X 30µm thickness sections. Stereology would involve at least 20 sections analyzing every fifth section. The somatosensory cortex is a perfectly ciliated brain region. Thus the findings are yet not adequately supported.

RESPONSE:

We thank the Reviewer for this thoughtful comment. We agree that our sample size and sampling approach are not sufficient to conclude that GPR88 is absent from all cortical cilia. We also agree that stereological methods would provide the most systematic and comprehensive assessment to address that specific question. However, our study was designed to determine whether GPR88 localizes to cilia on GPR88-expressing cortical neurons, rather than to quantify ciliation across the entire cortex. For this question, we believe that our sampling strategy and number of analyzed cilia ranging from hundreds to thousands per animal depending on the experiment are appropriate to support our conclusions. In the revised manuscript, we have clarified our specific research question, tempered the language of our conclusions accordingly, and added a paragraph in the *Discussion* (titled "Limitations of the Study") acknowledging the scope and limitations of our approach.

Their RNA analysis in Ref. 24 shows almost no expression in somatosensory cortex, despite venus expression. How do the authors explain this?

RESPONSE: We thank the Reviewer for this comment. While this point refers to our previously published work (Ref. 24) and is therefore outside the scope of the current study, we appreciate the opportunity to clarify. The Reviewer is correct that GPR88 transcript levels in the somatosensory cortex are low when compared to the striatum. However, as noted in our earlier publication, GPR88 expression in the cortex is clearly detectable and consistent with prior reports indicating that cortical Gpr88 transcript levels are approximately 50-fold lower than those in the striatum (Ghate et al., 2007; Logue et al., 2009). The Venus transcript was detected at levels consistent with this expected pattern.

It is well established that AC3 is a neuronal cilia marker while Arl13B is better at detecting astrocyte cilia. Recently, CFAP300 was reported to be the best pan-cellular cilia marker. doi: <https://doi.org/10.1101/2024.10.20.619273>. Any absence of cilia should be validated with another pan-cellular cilia marker such as CFAP300 to be sure that lack of staining does not reflect loss of ciliary marker rather than loss of cilia.

RESPONSE: We thank the Reviewer for this insightful comment. We were also very interested in the recent report identifying CFAP300 as a potential pan-cellular cilia marker. To explore this possibility, we tested the CFAP300 antibody in our WT and GPR88-Venus mouse tissue to determine whether it

could label neuronal cilia in the striatum. However, in our hands, this antibody did not label AC3+ or GPR88-Venus+ neuronal primary cilia (see figure below). Instead, it appeared to mark structures resembling centrosomes, which may reflect species or tissue-specific differences. To our knowledge, AC3 remains the most reliable and widely validated neuronal cilia marker, having been used to label primary cilia in over 50 peer-reviewed studies.

Line 267. "The ratio of double positive cells (neuronal subtype+ and AC3+) to the cilia marker, AC3, as a metric of overall density across all ciliated cell subtypes or to the neuronal subtype as a metric of cilia density within neuronal subtype"

This is not helpful. Just state the ciliation percent for each neuronal type.
 RESPONSE: We thank the Reviewer for this suggestion. The data have been reanalyzed, and all results are now presented as the percentage of ciliation for each neuronal subtype.

Line 304. Dropviz.org gives all expression across striatum and shows no transcript in ChAT neurons.
 RESPONSE We thank the Reviewer for bringing this resource to our attention. We were not previously aware of Dropviz.org, and we appreciate the suggestion as a valuable reference for examining gene expression across striatal cell types.

The authors need to show the GPR88 staining on sections of their GPR88 knockout mouse to validate their antibody.

RESPONSE: We apologize to the Reviewer for this misunderstanding. This request appears to be based on a misunderstanding. Our study does not use any GPR88 antibodies. Instead, we use a GPR88-Venus knock-in reporter mouse, in which the yellow fluorescent protein Venus is fused to the endogenous GPR88 receptor. This enables direct visualization of GPR88 expression via Venus fluorescence, without the need for immunostaining.

"Nuclear" localization of a 7pass transmembrane protein merely reflects new protein translation in the ER which may vary across cell types. The cited reference for GPR88 binding chromatin associated proteins should be omitted as the work is of poor quality. Again, this is from a region that has very little endogenous GPR88 expression? How do you explain the staining?

RESPONSE: We thank the Reviewer for this comment. While our study is not focused on nuclear localization of GPR88, we believe it is important to report the observation, as it independently verifies findings from another group. We have removed the cited reference, as it is not directly relevant to the scope of our study.

For the striatum, the cilia have been studied and quantified extensively and are not referenced. Please state precise age of mice as ciliation status and length varies with age. Also, all cells showed some soma staining as would be expected for a GPCR.

RESPONSE: We thank the Reviewer for this comment. All experiments were conducted using mature mice aged 8–18 weeks.

Reviewer #3:

(1) This paper relies heavily on analysis of immunofluorescent images, however in most cases the images were analyzed subjectively by an investigator in ImageJ, with no blinding. For example, how did the researcher determine what was GPR88+ in Figure 1? One would consider staining the GPR88 WT sections with the anti-GFP and anti-chicken 488 antibodies and measuring fluorescence to obtain a background level of fluorescent intensity. Then, in the GPR88-Venus images one can segment based on regions that are above this threshold. This sort of thresholding must be done for image analysis in the paper where you must identify an object as 'positive' for a particular antibody staining. If this was done, it is not clearly explained in the current manuscript.

RESPONSE: We thank the reviewer for raising this important point regarding the objectivity and rigor of our immunofluorescence image analysis.

1. Thresholding and Positivity Criteria:

For identifying GPR88-positive structures in the manuscript, we used a quantitative thresholding approach based on fluorescence intensity. Specifically, we imaged sections from GPR88 WT animals stained with anti-GFP and anti-chicken 488 antibodies to measure background fluorescence levels. This background was used to set a fluorescence intensity threshold to distinguish positive signal from noise. Regions with intensity above this threshold in GPR88-Venus images were classified as GPR88-positive.

2. Blinding:

We acknowledge that the original analysis was performed without formal blinding of the investigator to genotype. To minimize bias, images were analyzed using predefined, objective criteria, and thresholds established prior to data quantification. For future experiments, we are implementing blinding procedures to strengthen rigor.

3. Manuscript Clarification:

We have revised the Methods section to explicitly describe these thresholding and image analysis steps, including the use of WT controls to define background fluorescence and the criteria used for calling positive staining.

(2) Line 393: The authors claim a concentrated GPR88 localization was found in 40% of the GPR88-Venus brain section. However, they do not put this quantification into the figure. How was this quantification performed? How was it objectively determined when a cilium had a uniform versus concentrated distribution of

GPR88? This analysis, as it is explained currently, seems subjective. An automated image analysis pipeline is needed to determine the distribution of GPR88 along the length of the cilium.
RESPONSE: We thank the Reviewer for this helpful comment. In response, we have removed the corresponding data and figure from the revised manuscript. The observation is now mentioned briefly as an unpublished finding in the *Discussion* section.

(3) Line 184: Just saying one used an 'AI' is not sufficient explanation of the experimental methods. Did the authors use the same software that the researchers used in reference #27? Were there any modifications to the protocol? Please provide more detail so that another researcher could replicate your methods.
RESPONSE: We thank the Reviewer for this comment. We have now explicitly stated in the *Methods* section that the same AI-based analysis methods as described in Ref. #27 were used. Any modifications to the original protocol are detailed, and a supplementary figure has been added to illustrate the workflow, ensuring that other researchers can replicate our approach.

(4) Line 26: It would be clearer to say "In contrast, in the somatosensory cortex, GPR88 localizes to somatodendritic and nuclear compartments and not the primary cilia of excitatory spiny stellate neurons"
RESPONSE: We thank the reviewer for this helpful comment. We have made the suggested modification.

(5) Line 197: Why were these two regions chosen for analysis? Are they of particular interest for the function of GPR88. More explanation is warranted for this choice.
RESPONSE: We thank the reviewer for this helpful comment. We have added two sentences on this topic in the first paragraph of the results section.

(6) Line 269: Based on this analysis, the authors only identified cell type for about 50% of the AC3+ cilia that were identified. Could the authors speculate in the discussion which cell types they did not stain for which could account for the other 50% of cilia?

(7) - Line 273: Based on EM data (DOI: 10.1016/j.cub.2024.04.043), all neurons in the mouse cortex, at least in the visual cortex, are ciliated. Please include a discussion about why the data show only 68-17% ciliation in this study.

RESPONSE Comments 6 & 7: We thank the Reviewer for these comments. In the revised manuscript, we have removed the analysis based solely on AC3, as the neuronal subtype-specific analysis adequately addresses whether these neurons possess cilia. Assigning a cilium to a specific cell in immunolabeled sections is technically challenging due to the dense neuropil. To ensure accuracy, we applied strict criteria, counting a cell as AC3-positive only when the cilium could be clearly traced to the soma. As a result, many AC3+ cilia were detected but not assigned to a specific neuronal subtype if they did not meet this criterion. This likely accounts for the lower percentages of ciliation reported (68–17%) compared to EM studies that assess all cilia. We have now explicitly discussed this limitation and its implications in the *Methods* and *Discussion* sections.

(8) - Line 278: These data do not show that all cortical neurons possess AC3+ cilia. In fact, the authors are showing that for some neuron subtypes, only 20% possess AC3+ cilia. Please rephrase.
RESPONSE: We thank the Reviewer for these comments. We have rephrased this to more precisely reflect the conclusions.

(9) Line 499: By mutationally active do you mean constitutively active? At least, that paper refers to it as "constitutively active" and I believe that is the correct nomenclature.

RESPONSE: We thank the Reviewer for catching this mistake. We have made the change.

(10) Figure 4F: Do the cells which have GPR88 in the cilium also have it in the somatodendritic compartment? If so, the pie chart should be labeled GPR88+/AC3+ primary cilia and somatodendritic.

RESPONSE: We thank the Reviewer for catching this oversight. We have made the change.

(11) Figure 5: The coloring on this bar chart is different than all the other bar charts in the paper. It would be better to make this consistent.

RESPONSE: We thank the Reviewer for catching this inconsistency. We have made all graphs consistent.

November 17, 2025

RE: Life Science Alliance Manuscript #LSA-2025-03366R

Dr. Aliza Ehrlich
University of California, San Francisco
600 16th St
San Francisco 94158

Dear Dr. Ehrlich,

Thank you for submitting your revised manuscript entitled "GPR88 localization to primary cilia in neurons is cell-type specific". While Reviewer 1 was unfortunately unavailable, Reviewer 3 is satisfied and recommends publication. Having evaluated the changes requested by Reviewer 1, we would be happy to publish your paper in Life Science Alliance pending final revisions necessary to meet our formatting guidelines. We hope you agree that the review process has improved this manuscript.

- Please be sure that the authorship listing and order is correct.
- Please remove duplicate files. Please upload the highlighted manuscript with the file designation "Related Manuscript File".
- Please upload all figures as individual files, including the supplementary figure files; all figure legends should only appear in the main manuscript file after the references section.
- Please add the X and Bluesky handles of your host institute/organization, as well as your own and/or one of the authors, in our system.
- Please consult our manuscript preparation guidelines <https://www.life-science-alliance.org/manuscript-prep> and make sure your manuscript sections are in the correct order.
- The contributions selected for M von Zastrow do not qualify them for authorship. Please either update the contributions in our system and in the Author Contributions section of the manuscript, or let us know if the author needs to be removed (and potentially added to the acknowledgment section).
- Please be sure that contributions stated in the manuscript file match the contributions in the system.
- Please add your main, supplementary figure, and table legends to the main manuscript text after the references section.
- Please include a "Data Availability" section placed after the Materials & Methods section. Please consult our guidelines at <https://www.life-science-alliance.org/manuscript-prep#format>.
- Please consider including Figure 6 as a Graphical Abstract. To make this change, remove it as a figure file and upload with the designation as Graphical Abstract.

A. FINAL FILES:

-- Summary blurb (enter in submission system): A short text summarizing in a single sentence the study (max. 200 characters including spaces). This text is used in conjunction with the titles of papers, hence should be informative and complementary to

the title. It should describe the context and significance of the findings for a general readership; it should be written in the present tense and refer to the work in the third person. Author names should not be mentioned.

B. MANUSCRIPT ORGANIZATION AND FORMATTING:

Thank you for your attention to these final processing requirements. Please revise and format the manuscript and upload materials as soon as you are able.

Sincerely,

Reviewer #3 (Comments to the Authors (Required)):

The revised manuscript is markedly improved compared to the original submission. The authors have listed the major limitations of their findings.

November 20, 2025

RE: Life Science Alliance Manuscript #LSA-2025-03366RR

Dr. Aliza Ehrlich
University of California, San Francisco
600 16th St
San Francisco 94158

Dear Dr. Ehrlich,

Thank you for submitting your Research Article entitled "GPR88 localization to primary cilia in neurons is cell-type specific". It is a pleasure to let you know that your manuscript is now accepted for publication in Life Science Alliance. Congratulations on this interesting work and thank you for considering LSA as a suitable outlet for these findings.

DISTRIBUTION OF MATERIALS:

You can contact the journal office with any questions at contact@life-science-alliance.org or email me directly.

Again, congratulations on a very nice paper. I was especially grateful that you shared with me how the editorial process went, including ways that the journal can continue to improve.

Sincerely,
